# A Move towards Developing Usable Climate Change Adaptation and Mitigation Services for the Agricultural Sector

Mokhele E. Moeletsi [1,2,3,*] and Mitsuru Tsubo [3]

1   Agricultural Research Council—Natural Resources and Engineering, Private Bag X79, Pretoria 0001, South Africa
2   Risk and Vulnerability Assessment Centre, University of Limpopo, Private Bag X1106, Sovenga 0727, South Africa
3   Arid Land Research Center, Tottori University, Tottori 680-0001, Japan; tsubo@tottori-u.ac.jp
*   Correspondence: moeletsim@arc.agric.za

**Abstract:** Dryland farming is at the center of increasing pressure to produce more food for the growing population in an environment that is highly variable and with high expectations for the standard of their production systems. While there is mounting pressure for increased productivity, the responsibility to protect the environment and diminish the agricultural sector's carbon footprint is receiving growing emphasis. Achieving these two goals calls for a consolidated effort to ensure that the scientific community and service providers partner with farmers to create a sustainable food production system that does not harm the environment. In this paper, we studied the nature of the services present in the market and identified ways that could be used to improve the climate services available to the agricultural sector. Important factors that could increase the usability of climate services include coproduction, context-specific information, innovation, demand-driven services, timeliness of services, highly applicable information, provision of services in the correct format, services that increase user experience, specificity of services to a locale, and services that are easily accessible.

**Keywords:** climate services; dryland farming; barriers to the development of services; main stages for new service adoption; climate variability





## 1. Introduction

We are living in an era where farmers not only need to alter their management to cope with climate variability but must also take into consideration the impact of their operations on the environment. There is also pressure to produce more because of the increasing demands for food caused by the rising global population. Thus, organizations mandated to provide services to farmers need to take all of these factors into consideration. Climate services provide farmers with tailored climate information that can be used to alleviate risks and reduce vulnerability to climate change [1]. Providing climate services entails combining raw climate data with other auxiliary information to produce customized products and services [2]. Alexander and Dessai [3] have described climate services as the delivery of processed climate information that is relevant to a particular decision-maker.

Farming communities are slowly embracing the importance of integrating climate information into their decision-making, but it is not easy for developers to produce products that can be integrated into the decision-making process [4]. Climate services can help create more value for agricultural communities by increasing productivity and reducing losses [5]. Even though coverage of climate services has increased recently, there are still numerous barriers that prevent some services from being used by end users [6]. Documented flaws of the current climate services include the inability to provide relevant information for users and the large uncertainties in climate forecasts at a number of timescales [7]. Suggestions for improving climate services have included the establishment of good relationships

between producers of information and end users; the coproduction of climate services; and the production of context-specific services. In recent times, there has been much focus on developing climate services that are tailored to users [1]. The rate of adoption of such services is still unclear, but the notion of shifting away from general services towards specialized services has a chance of increasing the odds of adoption. This makes the target group clear, and engagements can be focused on that group.

There is a great need for climate tools that have a local context in developing countries. Most of the climate services and tools originate in the USA and Europe. It is important that these tools be adapted for local use. Agricultural conditions in most developing countries in Africa are very different from those in industrialized countries. Basic environmental conditions, such as climate and soil, differ from one place to another. Most importantly, management practices differ because of resource availability and the scale of operations. A commercial farmer with over 200 ha of land and advanced agricultural machinery has different approaches than a subsistence farmer with less than 1 ha of land. It is important that all farmers be taken into consideration when climate services and tools are developed.

African agricultural production is extremely low compared with that in other regions, mainly because of the scale of productivity, access to resources, and basic knowledge of agriculture and climate-related risks [8]. This disparity calls for climate services that have the potential to improve productivity by providing climate information that can help farmers make informed decisions. According to Vaughan et al. [8], the improvement of climate services in Africa has the potential to help countries meet overall sustainability goals such as eradication of hunger, action in anticipation of climate change, and reduction in malnutrition. Climate services have historically been more geared towards reducing the impact of climate variability and climate change on the agricultural sector, but there is a greater need to consider the introduction of climate services that address the mitigation of climate change effects because food systems are the main source of greenhouse gas (GHG) emissions that cause global warming. In this study, we address these two distinct approaches and explore ways of blending them.

## 2. Methods

Much research in developed countries has focused on climate services; there has been little comparable research in Africa generally and South Africa particularly. We took inventory of the climate services that have been developed for the agricultural sector and either have been published or have been documented online. We investigated the nature of these services to highlight challenges and opportunities for enhancing climate adaptation services, climate mitigation services, or an integration thereof.

The paper utilized a four-step approach: step 1: identification of key keywords and research questions; step 2: selection criteria employed; step 3: analysis of case studies and literature; and lastly, step 4: provision of recommendations that would help improve climate services in agriculture.

We sourced publications on climate services from Google Scholar, Google, and Pro-Quest Dissertations and theses global using a number of keywords, including "climate services", "climate services for agriculture", climate services for farmers", "climate services for adaptation", climate services for mitigation", "climate services for South Africa", "optimal dissemination of products and services", "user-friendly services", "user-experience design", "barriers to adoption of new services", "principles of innovative climate services", "new product development stages" and "drivers of success in new service development". Over 100,000 articles were obtained from May 2021 to December 2023.

Papers were deemed relevant for further analysis if they met the following criteria: components of climate services were clearly used and described; weather information was somehow used to make decisions relevant to any of the economic sectors but with special attention to agriculture; climate mitigation was incorporated into decision-making based on several inputs; the climate services considered could be applicable to an African setting;

the content included developing products and services, useful or usable climate services, or agrometeorological services.

Best practices obtained from the literature were then used to propose ways to enhance climate services and key factors that could be considered in exploiting usable climate services. From the search engines, two examples of climate services were identified, representing climate services for adaptation and mitigation. We also proposed the development of more integrated climate services that combined both adaptation to and mitigation of climate change effects. We present a hypothetical example of the application of integrated climate services to dryland maize production. We identify barriers to developing climate services, and we discuss how those services might be improved. Lastly, we consider and discuss how the steps that we recommend could be followed to improve the adoption of climate services in the agricultural sector.

## 3. How to Develop Efficient and Usable Climate Services

Climate services tailored for the agricultural sector have not been successful in the market, as shown by low adoption rates. The main factors that have affected adoption rates include [9] the accuracy of the information provided; the legitimacy of the information; the timeliness and relevance of the information; the relationship between developers and users of the product or service; the format of the service; and the appropriateness of the dissemination channel. Some authors attribute this usability gap to the tendency to produce services that are imposed on the market by developers instead of services introduced to satisfy the needs of end users [3]. To bridge the usability gap of climate services in agriculture, most researchers have advocated for: (1) coproduction; (2) tailored or context-specific services; (3) innovative climate services; (4) demand-driven services; (5) services provided in a timely manner; (6) applicable services; (7) appropriately formatted services; (8) services that improve the user experience; (9) services that are location-specific; and (10) services that are highly accessible (Figure 1).

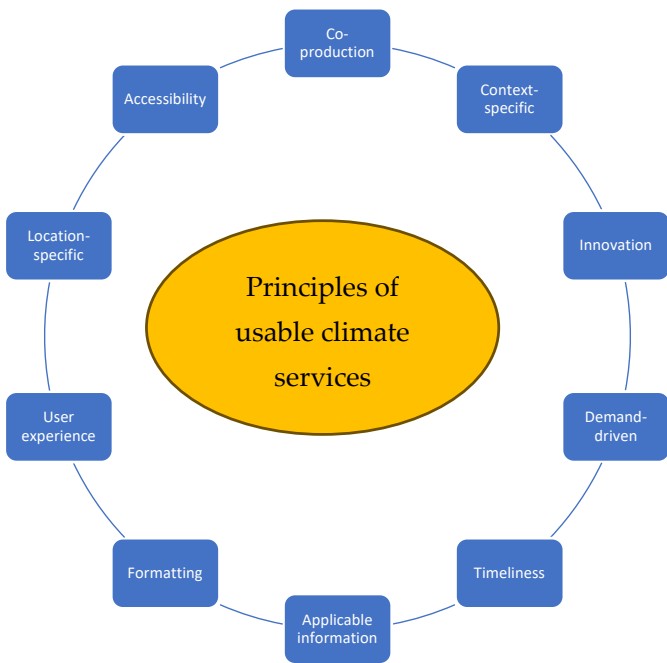

**Figure 1.** Components for developing usable climate services.

### 3.1. Coproduction

The low rate of adoption of climate services can also be attributed to a top-down approach wherein a service is developed with minimal user contribution. The concept of coproduction of services is regarded as key to increasing adoption. Coproduction is intended to shift the supplier-centric approach to the development of services towards a

more user-centric approach. It is described as a collaborative effort between the producers of the climate information and the intended users, with the aim of developing a service that is usable [6,10]. This scenario entails a genuine collaboration between developers of the services, intermediaries, and decision-makers on the ground [3]. It also means that scientists and end users work closely in planning, designing, and delivering services [11]. This collaboration has the potential to build trust between scientists and end users, which is a key ingredient for a successful service [12]. It also entails reframing questions for end users to ensure that they are defined well enough to enable easy interpretation by scientists or developers [7]. The clarity of what needs to be delivered is key to a successful service. This clarity will reduce the disjuncture between the expectations of end users and the services provided.

The involvement of intermediaries in the development and delivery of climate services has helped scientists decode requirements from end users [13]. In the past, this involvement was neglected, and there was little or no effort to ensure that information was transferred to end users in a way that facilitated its adoption. To improve cooperation and trust between scientists or developers and end users in coproduction, developers need to be aware of the constraints that can make it difficult to meet end-user requirements. For example, some of the information from climate models may not be at a fine enough scale to account for high spatial variability. The computational power of the system being used can also constrain the delivery of satisfactory services. To deepen the involvement of external stakeholders, Buontempo et al. [12] suggested the involvement of end users in the governance of climate services. The application of such an initiative could be tailored to particular operations, but the involvement of users in defining problems has been found to be important to closing the usability gap [12]. Numerous researchers have suggested a continual program of capacity building for end users in all stages of the development of climate services [7,12]. Such a program could also improve interactions between stakeholders, and if the interactions are intentional, the services will be well received by the end users.

### 3.2. Tailored/Context-Specific

The services provided to farmers need to be context-specific to have an impact on the ground [14]. The lack of tailored climate services is one of the factors that have contributed to the reduction in agricultural productivity in most developing countries [15]. End users have their own established ways of making decisions, and the one-size-fits-all approach will not work [13]. Climate information needs to be tailored for a specific user—the needs of the grain industry differ from those of the subtropical fruit industry, for example. While creating these tailored solutions, developers of climate services must take into consideration numerous factors, such as the risk profile of the commodity of interest. From an African perspective, climate services for agriculture are highly important due to their high contribution to GDP. Search results showed fewer case studies and low penetration of climate services in Africa than any other region, and this might be due to poor infrastructure, a lack of farm-level data, low literacy levels, a lack of expertise, inconsistent services, and a lack of access to capital [16,17].

### 3.3. Innovation

Innovation is important to developing appropriate climate services in the agricultural sector. Services that are developed mostly by government-funded organizations often become obsolete. This obsolescence could be due in part to a poor culture of innovation resulting from a lack of competitiveness among relevant organizations whose employees are not motivated to generate income for the survival of their institutions. Inquisitiveness is among the important attributes for developing services that can be ground-breaking in the marketplace (Figure 2). According to Collins and Lazier [18], ensuring that services are developed in a manner that is highly receptive to innovative ideas, regardless of whether the ideas originate internally or externally, can be key to developing sustainable climate services. Innovation in terms of the dissemination of climate services is also an essential

element that can help increase the usability of climate services, especially the use of social networks and social media platforms [19]. Several initiatives in the market are exploring the use of Fourth Industrial Revolution technologies to develop services that are relevant to users [20]. The success of such technologies in the development of climate services has not yet been apparent, but there is much scope for such technologies to improve the usability of climate tools and services.

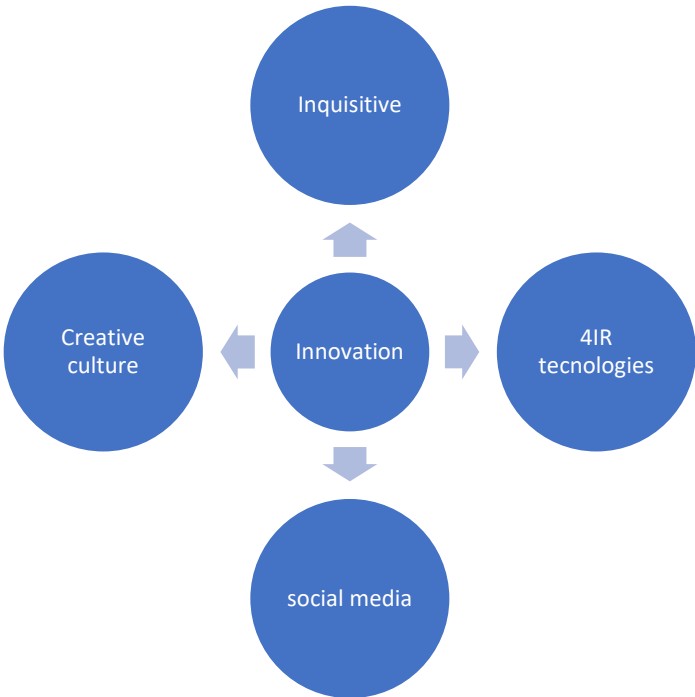

**Figure 2.** Some of the important elements of innovative climate services, including use of Fourth Industrial Revolution (4IR) technologies.

### 3.4. Demand-Driven

Climate services that are demand-driven stand a better chance of being used. Developers can produce demand-driven services only when they are in constant contact with end users. Being close to the customer has the advantage of making it possible to tune a product so that it resonates with market demand. Market research can facilitate the quest to become more relevant and to develop services that are usable. Some of the most successful services were produced through the identification of customer needs and efforts to meet them [18]. In contrast, as much as it is important for services to originate from consumer demand, some of the best products on the market were introduced into the market by developers. However, in such cases, there is a need for suppliers to engage end users to make them aware of the utility of the proposed service [13]. Some of these services may have been adopted from other industries or from different localities and, on the basis of their usefulness elsewhere, have been adapted to the new conditions that would make them appealing to customers. The need to strike a good balance between the two approaches is thus a key, and knowledge of when to give priority to one over the other is still unclear.

### 3.5. Timeliness

The timeliness of climate services is one of the main ingredients that can fuel adoption rates. Dissemination of the right information at the right time happens not by chance but rather because of the informed efforts of developers of products and services. A full understanding of the decision-making process of the targeted users is key to having a chance of developing climate services that are usable [1]. Farmers in most cases follow a distinct pattern when making decisions based on, inter alia, their experience, and indigenous knowledge. For example, when dealing with a maize farmer, service developers must

know the timing of activities such as preparing for ploughing, gathering inputs, and planting. They must also know the timing of distinct stages of crop development, especially the times of weeding, harvesting, and spraying of pesticides. Being able to adequately predict the timing of these activities can enable the dissemination of climate services that are relevant and the provision of recommendations tailored for the current stage of the crop [21].

### 3.6. Applicability

There are several great climate products that could be useful but are not currently used by end users. It is critical to ensure that the products that are developed are highly applicable to the needs of end users. Presenting information to end users without including its implications can be fruitless. Farmers need to understand the application of the information in their operating environment. Such understanding enables users to fit the information into their complex decision-making processes. Service providers should, however, not provide recommendations that are inaccurate and misleading because such information will harm productivity and lead to distrust of the services provided. It is better not to develop anything at all than to disseminate information that is highly uncertain. Research has shown that users would rather use their own instincts and knowledge than trust climate services that are poorly conceptualized [21]. Climate services based on course-resolution data do not normally make it to the decision-making process; the information may be deemed irrelevant because it is overgeneralized [1]. Another important factor is the provision of information well ahead of time when decision-makers need to adjust their management practices [15]. It is of no use to provide valuable information late because it has less applicability than timely information and may lead to a waste of resources. The use of terminology that is familiar to end users is also a key to the successful application of information. A common occurrence within the climate-service sector is the provision of services that are full of scientific jargon that is not adequately understood by the user community.

### 3.7. Formatting

The format of the climate service has been shown to play a significant role in the adoption rates. If it is not easy for end users to comprehend, it is highly likely that the service will not be used [10]. In Limpopo, we developed a coded short messaging system (SMS) that contained information on the possibility of a daily rainfall exceeding three thresholds: 1 mm (insignificant), 5 mm (moderate), and 20 mm (heavy) [21]. Even though we provided intense training and follow-up meetings to explain the service, farmers still struggled to interpret the messages. When developing services, consideration of the level of literacy and numeracy of the end users is crucial. Regardless of efforts by scientists, if the service is not presented in a way that users can comprehend, the adoption rate of the service will be low.

### 3.8. User Experience

We cannot over-emphasize the importance of developing user-friendly and engaging services that are designed for public consumption [22,23]. This development starts by creating an entry point to the service that is friendly, i.e., 'nice' to use by anyone. The rise of user-centric design has led to a shift from the user interface to the user experience (UX), where every component of the service is created with the goal of addressing users' needs and preferences [24]. Recent studies have advocated for user experience over user friendliness, even though some elements of the latter are present in the former [25–27]. The UX design of services focuses on having services with which interaction is enjoyable. It is important to shift from a supplier-driven paradigm to a market-driven paradigm wherein user experience is the key [25]. The current advances in computing that incorporate digital analytics can help analyze vast amounts of data available on the web that can improve users' satisfaction and interaction with our climate tools or products [25]. Climate service

developers need to find ways of incorporating their services into different social media platforms because more and more ordinary people enjoy using platforms such as Facebook and Tik Tok. This requirement calls for a different approach to product design and might require reskilling existing developers or hiring new digitally savvy developers.

*3.9. Accessibility*

The importance of the dissemination of climate information or of products or services to the farming community has been emphasized [28,29]. Most climate information does not reach the intended users and thus has little practical value, regardless of its theoretical utility [5]. The other crucial component of dissemination is the ease of access to information, which can be facilitated only by an efficient access mechanism [2]. Failure to use appropriate dissemination channels can greatly devalue a potentially useful climate service. Improvement of the dissemination of climate services is a goal for many organizations because of the usability of climate services around the world [19]. There are several ways to disseminate services to end users, including newsletters, television, radio, pamphlets, social media, computer applications, mobile applications, SMS, and social networks. These platforms have advantages and disadvantages that depend on the locality and resources available. Web services and social media have attracted a lot of attention in many countries. The use of social media platforms such as Facebook, WhatsApp, and X (Twitter) has grown by leaps and bounds, and many organizations are using these platforms to develop their services [30]. Even though an increasing number of people in rural areas have access to cell phones, web-based climate services still face constraints associated with unaffordable prices of mobile data that make access via cell phones unsuitable for resource-poor farming communities.

*3.10. Location-Specific*

The services provided to farmers need to be location-specific to stand a chance of helping farmers in their decision-making. There can likewise be important differences in the climate services that are provided. Consider a maize farmer in the eastern Free State of South Africa who has a narrow planting window and another one in Mpumalanga, where the challenges of frost are not severe. Because the timing of services for these farmers will differ, location-based climate services are needed. The challenge rests with the developers to ensure that local conditions are incorporated into the climate services. Current improvements in technology allow farming communities to have farm-level information. Mapping functionality can be used to pinpoint the location of a farm within the area of services being offered, and the specific information at that location can be sent to farmers at the required time intervals. This strategy can be implemented by using interpolation techniques or satellite-derived products to ensure that all areas are covered. The use of local language is also key in ensuring that climate services are location specific [31]. We cannot overemphasize the importance of indigenous knowledge, as it can be key to the adoption of climate services. Striking a balance between what is common knowledge in a specific area and scientific systems is highly recommended by other researchers.

## 4. Climate Services Available for Farmers

Climate services for agriculture can be categorized into those for adaptation to climate variability/change and those for climate change mitigation.

*4.1. Climate Services for Adaptation*

Climate variability is a major factor that affects agricultural production in most rainfed agricultural systems. A substantial body of research has convincingly documented that climate variability and extremes lower crop and livestock production potential in many regions. According to the World Meteorological Organization, up to 80% of the annual variability of crop yields is caused by climate, while climate hazards account for up to 10% of production losses [5]. Decreasing rainfall and extended drought in some regions

make it difficult to sustain traditional crop production. The distribution of rainfall over the growing period of most crops is erratic, especially in subtropical regions, and often it is not congruent with the water demands of crops. Large variability in the onset of the rainy season complicates planning for planting and increases weather-related risks towards harvest. The last two decades have been marked by increasing numbers of heat waves, sometimes with disastrous consequences for the production of both crops and animals. The productivity of rangelands has been affected by changing climatic conditions that have reduced the carrying capacity in some regions. Extremes in temperatures have also lowered livestock productivity in some instances.

There is an urgent need to reduce the negative impacts that climate variability has on agriculture [32]. Typical climate services are centered on the use of a climate monitoring network and weather forecasting. The decisions made by farmers, whether daily or on a seasonal timescale, are influenced by weather and climate [9]. Climate services are extremely important because they have the potential to provide valuable information about current agroclimatic risks and thus help farmers cope with climate variability [14,23]. Stigter [33] has suggested that some of the following climate services are key to agricultural productivity: climate advisories during the growing period; microclimate management as it relates to issues such as protection against damage by winds and frost; climate weather forecasts for agriculture; products and services for agroclimatic risk and characterization; early warning of pests and diseases; advisories for reducing the contribution of agriculture to climate change; and climate advisories for management of natural resources and grasslands. Many climate services and tools for coping with climate risks are available. This capability has been an area of focus for many organizations for a long time.

In this section, we will show an example of a service created for Australian farmers—My Climate View (https://myclimateview.com.au, accessed on 16 December 2023). This web-based tool was developed by the Commonwealth Scientific and Industrial Research Organisation and the Bureau of Meteorology to help farmers in Australia manage climate risks and make informed decisions [34]. The tool is based on past temperature and rainfall data obtained from weather stations all over Australia for more than 60 years, ERA5 reanalysis data on relative humidity, projected future climate data, past recorded or estimated soil moisture and potential evapotranspiration data, and seasonal forecasts of 1–3 months [34]. The information supports the majority of agricultural commodities in Australia, including subtropical fruits, grain crops, leguminous crops, deciduous fruits, citrus fruits, grapes, and livestock. Most importantly, the tool is coproduced with Australian farmers. The involvement of agricultural scientists, meteorologists, social scientists, marketers, and communication specialists with the tool has improved the understanding of the kind of information that could be useful to the Australian agricultural community. The tool is user-friendly and requires minimum information about the location and commodity of interest. The other tools that are available on the market to help farmers cope with climate variability and change are shown in Table 1.

**Table 1.** Other selected climate services for adaptation.

| Climate Service | Basic Functionalities | Applicability |
|---|---|---|
| Agroclimate (http://agroclimate.org, accessed on 16 December 2023) | - This tool can be used by farmers to manage climate risk, and it utilizes climate forecasts as well as recorded climate data to provide agricultural stakeholders with valuable information [35]. <br> - The tool has the functionality to monitor different parameters important to growers, like growing degree days, chill units, and heat stress. <br> - It can also help farmers with ENSO climatology and forecasting. <br> - It also has subsidiary tools for pests and diseases for selected crops. | Selected areas in the United States of America |

**Table 1.** *Cont.*

| Climate Service | Basic Functionalities | Applicability |
|---|---|---|
| CLIMTAG (https://climtag.vito.be/en, accessed on 16 December 2023) | - The tool equips agricultural stakeholders with climate information to help build climate-resilient agricultural systems [36]. <br> - The tool uses rainfall and other climate parameters from reanalysis data (ERA5) and climate projection data (CMIP5) to estimate a variety of agrometeorology indicators. | Africa |

## 4.2. Climate Services for Mitigation

Because of the requirement to feed the current global population of more than eight billion people, food systems are among the leading sectors of GHG emissions into the atmosphere [22]. Crop production, livestock production, forestry, and land use changes associated with agriculture are responsible for ~30% of total GHG emissions [37,38]. Studies have estimated that food systems globally contribute 25–34% of total GHG emissions [39–41]. There are major concerns that emissions from enteric fermentation are increasing with time in most countries, although there are a few exceptions in developed countries [42–45]. Emissions from manure management are a key contributor to elevated greenhouse warming [46]. Loss of soil carbon due to land use practices and expansion of crop cultivation are also major contributing factors [38]. Other sources of emissions include fertilizer use, and the intensification of agriculture has led to large increases in the application of nitrogen fertilizers [47].

Climate services are crucially important to ensuring that agriculture becomes a solution to climate change rather than a contributing factor. There is a need to reduce carbon emissions per food produced in all food systems. To achieve this major objective, a balance must be struck between productivity and the reduction in emissions. This balance must be determined at a point where agricultural activities yield optimum production at low levels of emissions. Scientists must push the limits in order to find solutions that would sustain production while at the same time reducing emissions. This might take time to realize, but a consolidated effort by all stakeholders can make it a reality.

Several tools and services have been developed in recent times to help agriculture and food production systems become more sustainable [22,38,48,49]. This section showcases Holos software v.4.0, which was developed for Canadian agriculture [38,50]. Holos (https://agriculture.canada.ca/en/agricultural-production/holos, accessed on 16 December 2023) estimates GHG emissions at the farm level for Canadian farmers [51]. Its aims are to find ways to reduce the carbon footprint of agricultural systems and to increase soil carbon sequestration through the alteration of agricultural management practices. The program covers emissions from cropping systems, enteric fermentation, manure management, land use change, and farm energy use. Holos uses the 2006 IPCC guidelines to estimate all the above-mentioned GHG sources [51,52]. Important input data include tillage systems, types and areas of crops planted, the number of livestock per livestock subcategory, the type of animal feeding, manure management systems, and amounts of energy used in farm operations. Kröbel et al. [53] explored the use of Holos software to quantify soil carbon change in a crop rotation system. Holos was used to identify the best animal feeding regime in a study that simulated strategies based on alfalfa silage and maize silage in a Canadian dairy production system using a life cycle assessment approach [54]. Holos has also been used in the assessment of cradle-to-farm-gate emissions of dairy and beef systems; the assessment of below-ground and above-ground carbon stocks of shelterbelts; the determination of GHG emission intensities of cow–calf beef operations across different climate regions; and the estimation of carbon footprints from high-grain-fed pig farms [54–60]. Table 2 shows some of the tools available on the market to estimate farm-level carbon emissions.

**Table 2.** Other selected climate services for mitigation.

| Climate Service | Basic Functionalities | Applicability |
|---|---|---|
| Cool farm tool (https://coolfarm.org/the-tool/, accessed on 16 December 2023) Started in 2008 | - It estimates farm emissions from both livestock and crop production using already established models from the Inter-governmental Panel on Climate Change (IPCC) national greenhouse gas inventory guidelines, independent research, and corporate greenhouse gas protocols [48]. <br> - Input data include area planted, types of crops, amount and type of fertilizer applied, type and amount of pesticides and herbicides applied, amount of electricity and other energy sources used, livestock population and type, and amount of feed and manure management systems. | Worldwide |
| Farm carbon Toolkit (https://farmcarbontoolkit.org.uk, accessed on 16 December 2023) Started in 2008 [61] | The tool can be used by farmers and agricultural stakeholders for all the major agricultural commodities, including livestock production, crop and horticulture production. <br> - The calculator covers scope 1, 2, and 3 calculations with a customized boundary, which can be (1) to the farm gate only; (2) farm and distribution, and (3) farm and distribution to the final customer [62]. <br> - The tool uses IPCC guidelines and other resources to estimate most of the greenhouse gas sources. | United Kingdom |

### 4.3. Integrated Climate Services for Adaptation and Mitigation

Traditional climate services have focused on building the resilience of end users by addressing the impacts of climate variability on sectors. Since 2000, growing numbers of climate services have been geared towards reducing GHG emissions from certain sectors, including agriculture. However, there is a need to combine these two related services into an integrated climate service that focuses on both climate adaptation and mitigation. This approach is critical to achieving most of the millennium goals and enabling a move towards climate action [2]. However, much remains to be done before we see a well-integrated climate service that incorporates both climate mitigation and adaptation strategies. Meteorologists and climatologists are not well trained to address the new demands for mitigation. The solution might rely on forging a collaboration between traditional climate scientists and sustainability practitioners to produce a well-balanced system that can enable farming communities to manage both risks.

Hypothetical Climate Service for Dryland Maize Production

The main goal of a perceived climate service is to enable optimal production by maize farmers while protecting the environment (Figure 3). The provision of climate information and modelling outputs will help farmers manage climate risks before planting, during the growing period, and after harvest to minimize crop losses and improve productivity. Algorithms need to be designed to enable a farmer to produce more maize with a lower carbon footprint. Climate services must have built-in memories to capture historical information about management practices and productivity outputs, such as yield and biomass, as well as the previous season's carbon footprint.

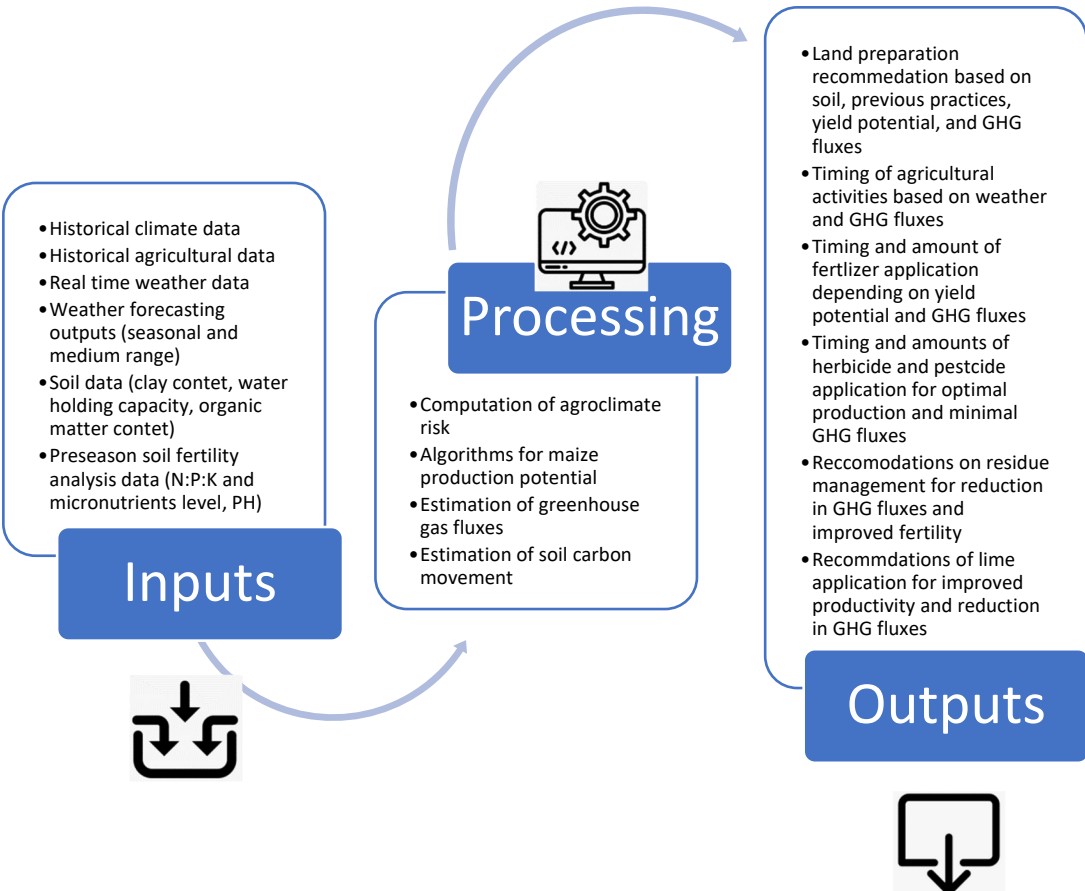

**Figure 3.** Hypothetical workflow of an integrated climate mitigation and adaptation service.

## 5. Barriers to the Adoption of New Climate Services

Many organizations and individuals have been continually developing services facilitated by the digital world, but such efforts are fruitless if the services are not adopted and used [63]. It is important for developers of climate services to understand that a new technology/service is not immediately embraced by the market, and efforts need to be created to break the common barriers to adoption. It is common knowledge that anything that threatens the status quo will face multiple barriers [64]. Several papers ([58,59,65]) have established that new products are faced with the following kinds of barriers: (a) technological; (b) financial; (c) organizational; (d) psychological; (e) functional; and (f) time (Figure 4).

(a) Technological barriers: Introducing a climate service to users can be challenging with respect to technology. A new service will normally be qualitatively compared with existing tools, and farming communities need to see considerable advancement in order to change or incorporate the new technology into their decision-making process [13].

(b) Financial barriers: Sometimes it is not feasible to conduct proper market research and follow fully cooperative principles because of limited financial resources [13]. There must be a cut-off in terms of how far some of the initial processes can go to ensure enough financial resources for the entire climate services project cycle. Many researchers are adamant that more financial resources are needed to move climate services from the category of being useful to being usable [13]. From the user's perspective, a new product needs to provide value for the money or time spent using it. Policies and frameworks need to be developed to enhance climate services that are subsidized for resource-poor farmers in an effort to ensure productivity at the small-scale level. Ensuring that the services provided are cost-effective is an ingredient that will increase the likelihood of adoption and higher market share. Profits in primary

agriculture are minimal and thus costing services for farmers need to be approached with care.

(c) Organizational barriers: It is very important that a proposed climate service be aligned with how operations work in order to avoid rejection. This requirement might necessitate a detailed study by the developers and intermediaries to understand the industry so that the climate service can be adopted immediately. The decision-making process of farmers and the agricultural community is complicated and involves input from multiple factors. A clear value addition is required for the use of a new resource to have a chance in the decision-making process. A number of capacity-development sessions with the users are required to break this barrier as well as to establish a client support system after evaluating a climate service. Organizational boundaries can also be bridged through the development of policies at national and local levels that enhance issues like cross-agency collaborations on matters that are central to the functioning of a society [66]. Policies are also integral to ensuring the penetration of services for beneficiaries by establishing enablers along the value chain.

(d) Psychological barriers: The conditioning of users of climate services varies as a function of their history, status, beliefs, religion, and financial situations. The adoption of a new service or product may be subjected to emotions of fear, anger, and disgust because of the potential of the new service to alter the routines of the end users [64]. In some cases, a fear that innovation will adversely affect the livelihoods of people can result in a lack of trust in the service. The attitudes of gatekeepers in some of the farming communities can play a role in the success or failure of a new service [63].

(e) Functional barriers: Users who are not eager to change their way of thinking and thus introduce a new climate service that functions unfamiliarly can create challenges for developers. It is important that the functionality of the new service be aligned with their current capabilities and technologies to lower the barrier to adoption [64]. Something new is normally perceived as complex. Factors such as farmers' profiles affect the adoption rate, and issues such as age and literacy can play a huge role [63]. This barrier can be lowered by dedicated client training and aftercare support.

(f) Time barriers: The development of climate services is highly time-consuming, and at times it is not easy to fully explore all the possibilities [13]. It takes a lot of time to convince users to utilize a service. In some cases, users are not prepared to explore innovative solutions because of the initial learning curve.

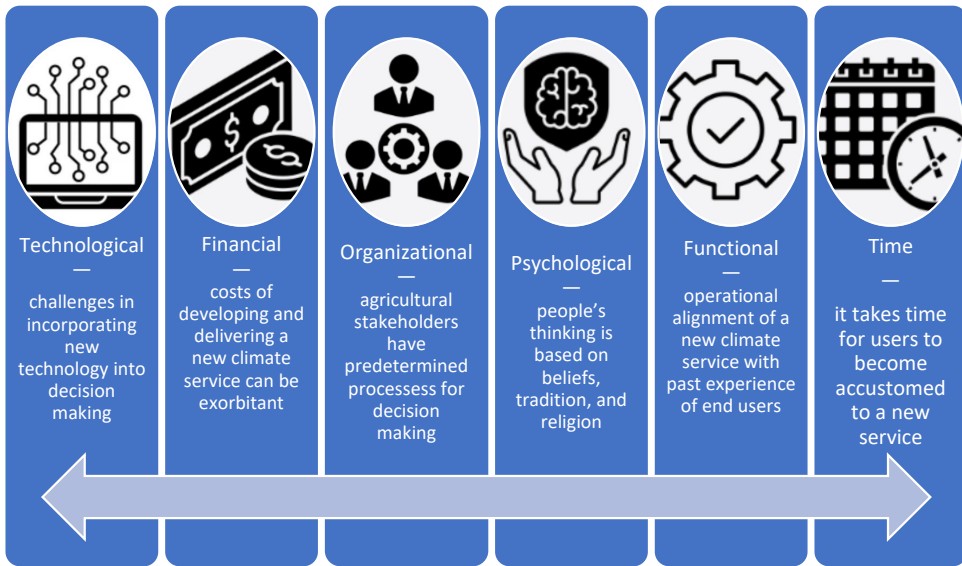

**Figure 4.** Barriers to adoption of a new climate service.

**6. New Climate Services Development Stages**

It is important to ensure that a new climate service or tool has an increased chance of success in the market. Organizations should not leave the success of a new product to chance but rather should take the necessary, proven steps to increase the likelihood of its adoption. Many climate services have died along the way because they were not meticulously planned. Proper time management is required for the development and delivery of climate services, partly because of the limited number of personnel to do the work. Persistence and continual upgrading of the service are required as the relationship with users is built. To create a sustainable service, it might be necessary for climate service developers to partner with another institution that is mandated to operationalize climate services. We believe that a climate service for any agricultural commodity needs to be provided indefinitely because it fills a vacuum in decision-making. This mindset will reduce the number of discontinued climate services.

There are five main stages for product or service adoption [67,68], namely (a) awareness stage; (b) research and discovery stage; (c) evaluation stage; (d) trial stage; and (e) adoption or rejection stage.

(a) Awareness stage: It is important to ensure that the climate service introduced into the market is visible at important information points. A summary of the product can be published in popular agricultural magazines or newsletters to increase its visibility. In most countries, there are several important agricultural shows during the year where the services of the new product can be presented. Strongly marketing the new service is paramount to market penetration and increases the chances of its being used. In the awareness stage, introducing capacity-building programs for farmers and other major stakeholders is key to improving adoption rates. Capacity-building initiatives need to be incorporated throughout the lifecycle of the climate service.

(b) Research and discovery stage: End users need to have confidence in the climate services being offered [13]. This confidence can be bolstered by demonstrating that the service is an improvement over existing services or by introducing valuable information that is deemed useful by the stakeholders. Most climate services do not go past this stage, dying a natural death before being used by farmers. Ensuring that end users understand the value that a new climate service brings is important in the second stage of development. In this stage, any feedback mechanisms that can improve the usefulness of the product are encouraged. This stage involves much capacity-building among the users, with an important focus on major informants and potential early adopters who are influential in the agricultural sector.

(c) Evaluation stage: This is a crucial step where users compare the proposed climate service with other products on the market. If there are no competing products, potential clients gauge the usefulness of the service and whether to use it. Its functionality and ease of use will encourage early adopters to take a chance with the product.

(d) Trial stage: All users require a trial period to decide how well a product addresses their real-time challenges. It is important to provide as much time as possible for people to test a climate service. Increasing contact with potential end users and incorporating their views on how to improve the service is needed to sustain the service. It is important to ensure minimal barriers to adoption at this stage.

(e) Adoption or rejection stage: The inevitable reality after some time has passed is that the new climate service will be adopted or rejected by the targeted group. The adoption of the service will be an indication that the initial marketing, functionality, and usefulness of the service were apparent to the users. It is important to further engage users in sustained use of the service. Continual upgrading of the services and support of the users is always crucial at the adoption stage. Thus, incorporating a monitoring and evaluation (M&E) procedure from the inception of the climate service project to the full deployment of the service is important. M&E needs to cover all aspects of the climate service. Any lapse can result in the defaulting of clients. It is

important to ensure that the necessary financial support, human capital, and other resources that are needed to support the service are continually available.

## 7. Conclusions

We examined how providers of climate services could develop products and services that are usable by farmers and could help farmers produce food profitably. Climate services for the agricultural sector can take the form of climate change adaptation, climate change mitigation, or a combination of the two. Traditional climate services have been based only on addressing climate variability and the impacts that climate change has on agricultural systems. The importance of climate change mitigation has been highlighted in many studies and presents opportunities for the abatement of harmful GHG emissions from the agricultural sector. We showcased some of the applications in agriculture that reduce carbon footprints and help farmers cope with climate variability and risk. We also demonstrated the importance of transitioning towards an integration of services for adaptation to and mitigation of climate change in order to increase the resilience of farming systems to climate variability and change.

Key barriers to the adoption of climate services—technological, financial, organizational, psychological, functional, and time—need to be addressed. It is also important for developers of climate services to follow and explore all recommendations of established processes for the introduction of any new invention (awareness stage, research and discovery stage, evaluation stage, trial stage, and adoption stage) into the market.

This paper presents some of the factors that can help organizations transition from developing useful services that are supported by highly credible science to making usable services that are ready to be incorporated into the decision-making processes of clients.

**Author Contributions:** Conceptualization—M.E.M.; methodology—M.E.M. and M.T.; undertaking of research—M.E.M. and M.T.; writing—M.E.M. and M.T. All authors have read and agreed to the published version of the manuscript.

**Funding:** This research received external funding from the National Research Foundation in South Africa and the Japan Science and Technology Agency under their Africa-Japan Collaborative Research program, grant agreement No. 132804 and SICORP grant No. JPMJSC20A4.

**Data Availability Statement:** No new data were created or analyzed in this study. Data sharing is not applicable to this article.

**Acknowledgments:** We thank Agricultural Research Council colleagues for editing the initial paper.

**Conflicts of Interest:** The authors declare no conflicts of interest.

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
