# Peer review of "A Move towards Developing Usable Climate Change Adaptation and Mitigation Services for the Agricultural Sector"

_climate, doi:10.3390/cli12030046_

Round 1

Reviewer 1 Report

Comments and Suggestions for Authors

The authors of the paper "Challenges and Opportunities for Climate Services in the Agricultural Sector" present a perspective on the challenges and opportunities related to climate services for the agricultural sector. They emphasize the need for integrated climate services that address both adaptation and mitigation strategies, and they highlight the importance of user experience and usability in service design. The paper also advocates for co-production, tailored services, innovation, and demand-driven approaches to improve the usability and adoption of climate services. While the paper could benefit from more empirical research and methodological clarity, it offers a unique perspective on addressing the challenges faced by the agricultural sector in the context of climate change. The detailed review is attached herewith. 

Comments on the Quality of English Language

Minor editing of English language required

Author Response

See attached responses to the reviewer's comments

Reviewer 2 Report

Comments and Suggestions for Authors

 The paper is of high significance for the scientific communities to utilize the findings as per their circumstances. 

Congratulations!

Comments on the Quality of English Language

Authors need to go through the sentence structure and flow of writing.

Author Response

We have addressed the reviewer's comment from the attached document

Reviewer 3 Report

Comments and Suggestions for Authors

This paper is well-written and clearly presented.  The summary of the many issues facing the commercialization of research into practical services will be useful for researchers considering how to translate results into tools for producers.  Overall it is a thorough and accessible summary of the challenges faced in developing services.  The one issue that I thought could maybe be addressed in more detail is, in considering cost of the service, that producers need to feel reasonably certain that their financial gain from using a resource will exceed the cost of the service.  In my experience this has been the most significant issue (in the Canadian context), where, for example, expensive services that work well for very large scale US operations are too expensive relative to any potential benefit for smaller scale farms. 

Author Response

We have addressed the reviewer's comments using the attached documents

Round 2

Reviewer 1 Report

Comments and Suggestions for Authors

As a perspective article, the manuscript is well structured. I thank the authors for incorporating a few changes (tables & statements) in their article. However, certain aspects need to be addressed (attached file) before the manuscript is accepted. 

Comments on the Quality of English Language

Minor editing of English language required

Author Response

Dear Editor

We have addressed all the reviewer's comments.

Looking forward to your further communication

Kind Regards

Mokhele
